# Forensic Hydrology Reveals Why Groundwater Tables in The Province of Noord Brabant (The Netherlands) Dropped More Than Expected

**Jan-Philip M. Witte [1],\* , Willem Jan Zaadnoordijk [2] and Jan Jaap Buyse [3]**

[1]  KWR Watercycle Research Institute, Groningenhaven 7, 3433PE Nieuwegein, The Netherlands & VU University Amsterdam, Department of Ecological Science, De Boelelaan 1085, 1081HV Amsterdam, The Netherlands

[2]  Geomodelling, TNO Geological Survey of the Netherlands, Princetonlaan 6, 3584 CB Utrecht, The Netherlands & Water Resources Section, Faculty of Civil Engineering and Geosciences, Delft University of Technology, Stevinweg 1, 2628CN Delft, The Netherlands; willem_jan.zaadnoordijk@tno.nl

[3]  Vitens N.V., Oude Veerweg 1, 8019BE Zwolle, The Netherlands; janjaap.buyse@vitens.nl

\*  Correspondence: flip.witte@ecohydrology.nl; Tel.: +31-612-137-751

**Abstract:** Since the nineteen fifties, groundwater levels in the Netherlands dropped more than as simulated by hydrological models. In the rural sandy part of the Netherlands, the difference amounts to approximately 0.3 m on average. The answer to the question of what or who caused this 'background decline' of groundwater tables may have juridical and financial consequences, especially since Dutch farmers are entitled to financial compensation for crop damage caused by groundwater abstractions. In our forensic study, we investigated how anthropogenic changes in groundwater recharge from 1950 to 2010 affected groundwater levels. In this period, crop yields in agriculture have risen sharply, and, because crop water use is proportionate to crop production, this led to more crop evapotranspiration and subsequently less groundwater recharge. Urban expansion and forestation has also led to a decrease in groundwater recharge. We showed that these changes in recharge may have caused a decline of groundwater of 0.2–0.3 m over 60 years (1950–2010). The simulated drawdown caused by groundwater abstractions appeared to depend on the amount of groundwater recharge related to land use and crop yield. This means that to properly evaluate the effects of a particular groundwater abstraction, one should account for the hydrological history of the landscape since the start of that abstraction.

**Keywords:** crop yield; groundwater abstraction; groundwater recharge; history; land-use; urbanization

## 1. Introduction

Water scarcity is becoming an increasing problem worldwide, especially in arid regions. A lack of fresh water has been considered a serious cause of current and future conflicts both regionally and internationally [1,2]. In humid regions, the problems are less socially and politically severe. As a result, publications about regional water scarcity in such regions are scarce. However, in humid regions, scarcity does also arise during parts of the year because the water demand is temporarily larger than the available amount of water which can be used without causing undesirable effects.

It may seem odd that in the Netherlands, water scarcity caused a conflict, since this country (35,000 km$^2$, Figure 1) is situated in the delta of the rivers Rhine and Meuse and because it has a humid climate. Moreover, groundwater levels in most of the country are shallow (less than 1–2 m below soil surface), so that in the growing season (summer), capillary flow from the groundwater table to the rooting zone is an important source of water supply to plants [3]. Why then, with this apparent

surplus of fresh water, should there be water scarcity? The answer lies in the fact that water tables in The Netherlands are controlled as much as possible to meet the demands of land-use functions, particularly of agriculture.

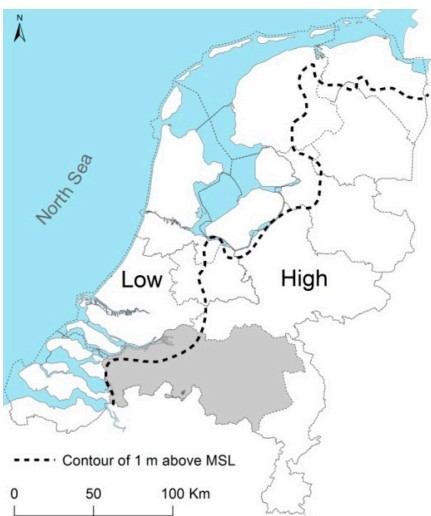

**Figure 1.** Provinces of the Netherlands (grey: Noord-Brabant) and division of the country in a relatively high part and a low part.

After World War II, Europe developed a common agriculture policy to abandon hunger and starvation [4]. In line with that, the Dutch government stimulated land improvement to increase crop production. The Netherlands succeeded like no other country, with the result that, nowadays, after the United States, it is the second largest exporter of agricultural products in the world (www.cbs.nl/en-gb). To achieve this, in most areas, the water management system was considerably adapted, especially to get rid of excess water, by digging and re-dimensioning ditches and by installing drainage pipes, weirs, and pumps. As a result, groundwater levels dropped and crop production increased, as intended. Though this operation brought wealth to farmers and the rest of the Dutch population, there were also unintended negative side-effects: heavy pollution of soil, water and air [5–7], and decline of groundwater levels in nature areas adjacent to farmland, which resulted in a serious loss of conservation values. In 1990, the Dutch parliament recognized this loss as a national problem that should be tackled [8]. The basis for this decision was a number of research reports, among which was the analysis by Rolf [9] of trends of groundwater levels observed from 1950 till 1989 in a large number of piezometers installed outside the influence of groundwater abstraction wells in the sandy Eastern and Southern 'high' part of the country (Figure 1). Rolf [9] found that groundwater levels in these four decades had dropped, on average, 0.3 m more than he could explain using a time series of precipitation and reference evaporation. This gap of 0.3 m he called 'achtergrondsverdroging', which can be translated as 'background decline'. A background decline of similar magnitude was also observed by others who used numerical or analytical groundwater models. Since the publication of Rolf [9], numerous reports and papers appeared about this hydrological black hole, unfortunately all in Dutch (see Supplementary Material): Hydrologists analyzed groundwater level time series in a different manner, used additional information like soil maps, and tried to find flaws in computation methods and in measurement techniques, etc. In the last decade, the debate among Dutch scientists intensified, and in 2013, the association of Dutch hydrologists started a study commission on 'background decline'. In 2016, the editorial staff of the most important professional journal for Dutch hydrologists, *Stromingen*, announced a complete ban on the matter: Publications about background decline would no longer be accepted as further discussion on the topic would not provide new insights.

The debate about the causes of the background decline is not only of interest to hydrologists but might also have serious juridical and financial consequences, especially because Dutch farmers are

entitled to financial compensation for crop damage caused by groundwater abstractions. Because of this, drinking water companies pay millions of euros to farmers every year. More expensive but harder to quantify are the salaries of a whole army of civil servants, lawyers, judges, specialists, and arbitration commissions who are involved in disputes over the effects of groundwater extractions on crop yields.

There have been many changes since 1950 that could have caused the background decline of groundwater levels in the Netherlands. The history of the Dutch landscape needs to be considered to study these causes, since many landscape changes have hydrological consequences: Wet areas have been extensively drained; cities have expanded; abstraction wells were installed for drinking water supply, industry, and agriculture; infrastructure was constructed, sometimes disrupting impermeable layers; polders were reclaimed from the sea and lakes; in clay and peat areas, drainage caused considerable soil subsidence; etcetera. Without careful investigation of these causes, the measured decline in groundwater levels cannot be explained or is wrongly fully attributed to the only causes considered.

In this paper, we will focus on one possible cause of background decline not considered in previous studies, namely the anthropogenic changes in the groundwater recharge since 1950. Groundwater recharge is defined here as the amount of precipitation that does not evaporate, run off superficially, or disappear in the sewer system, but eventually percolates to the groundwater. Crop yields in agriculture have risen sharply over the last half century, and because water use of crops is proportionate to crop production, this must have led to more crop evapotranspiration. In addition, grasses, shrubs, and trees became more abundant in nature areas, partly under the influence of atmospheric nitrogen deposition and cuts on nature management. Those plants evaporate more than the original vegetation. Urban expansion, too, may have contributed to the reduction of groundwater recharge due to the fact that in urban areas a large part of the precipitation water flows into the sewer system and no longer reaches the groundwater.

Our study is an example of 'forensic hydrology', a term that was introduced in the scientific community in 2007 by the Southwest Hydrology magazine. In the preface on a volume dedicated to the topic, the publisher wrote [10]: "Southwest Hydrology's approach is to look at the hydrologic tools available to determine the history of an event—such as water contamination, recharge, or groundwater capture—that matters to some entity, for example, a manufacturer, well owner, or municipality. This issue's authors also touch upon ways to make forensic investigations successful in and out of the courtroom".

In our forensic study, we will analyze how changes in land use and in crop yield have affected groundwater recharge and groundwater table. We focus on changes in groundwater recharge and will not quantify the possible contribution of other changes in the landscape to the background decline of the groundwater table. We will compare two historical periods: the period around 1950 and around 2010 (i.e., 1947–1953 and 2007–2013, respectively). The reason for this approximate time denotation is that some of the available historical statistics cannot be attributed to a particular year. For both periods, further referred to as 1950 and 2010, we will investigate both land use and the yields that were achieved in agriculture. We will demonstrate that these historical changes had an impact on the groundwater table, which cannot be neglected. We will focus on the province of Noord-Brabant (Figure 1), but the results of our study are applicable to the entire high part of the Netherlands, which mainly consists of Pleistocene cover-sand, intersected by brooks. Dominant land-use is agriculture (especially dairy farming and bio industry).

## 2. Materials and Methods

### 2.1. General Approach

We used published figures about land use and annual crop yields (Section 2.2) to estimate changes, between 1950 and 2010, in annual groundwater recharge. Since surface-runoff hardly occurs in the Netherlands, we assumed groundwater recharge on average equals precipitation (*P*)

minus actual evapotranspiration (*E*). Evapotranspiration consists of three components: transpiration, soil evaporation, and evaporation from wet surfaces during and after a precipitation event (interception). We transferred crop yields into annual transpiration figures, but to estimate the annual evaporation from bare soil and from wet surfaces, two very dynamic processes, we made use of a dynamic one-dimensional vadoze zone model, running with a time-step of 1 d (Section 2.3). This vadoze zone model was also needed to scale annual evapotranspiration figures down to a time-step of 5 d, the temporal resolution of a groundwater model we used to assess the effects on groundwater levels (Section 2.4). Groundwater recharge was imposed on this groundwater model as an upper boundary condition, thus deliberately disregarding soil water uptake by plants which, after all, was already accounted for in the estimate of actual evapotranspiration from crop yields.

For our calculations, we used daily precipitation and reference crop evapotranspiration data in 2007–2013 from the Royal Netherlands Meteorological Institute (KNMI, http://www.knmi.nl). By doing so, the differences in the computed groundwater recharge are only due to changes in land use and crop yield (the focus of our study), and not to differences in weather conditions between 1950 and 2010. For 2007–2013 the average precipitation in the province of Noord-Brabant was 831 mm·y$^{-1}$ and the average reference evapotranspiration according to Makkink [11] ($E_{ref}$) was 601 mm·y$^{-1}$. In Section 2.3.3, we will estimate the groundwater recharge of arable land in 1950 and 2010 from published figures about crop yield in both periods and precipitation values of 2007–2013. Since crop yield is affected by weather, this estimation is only valid if the growing seasons of both periods had comparable meteorological conditions. Indeed, the Dutch climate has changed from 1950 to 2010, but this change is especially apparent in more precipitation in winter, while *P* and $E_{ref}$ in the growing season hardly changed [12]. More importantly, both periods (1947–1953 and 2007–2013) had a comparable 'maximum cumulative precipitation deficit' in the growing season (max $\sum(E_{ref} - P)$ from 1 April to 30 September) [13], a measure which correlates well with observed yield figures [14]. For this reason, we believe it is acceptable to compare groundwater recharge rates in 1950 and 2010 of arable land on the basis of crop yields.

*2.2. Changes in Land Use and Crop Yield*

Both main categories of land use (i.e., 'forest and nature', 'urban area' (built-up area), 'arable land', and 'other') and crop yields were derived from the only source available for this purpose: Statistics Netherlands (CBS, www.cbs.nl/en-gb/). For crop yields, CBS statistics issued figures for the nine regions within the province that are depicted in Figure 2. These regions have been defined for administrative reasons only; each region comprises a number of municipalities and one central nucleus (usually a city) with an important regional function.

Throughout the province, the urban area has grown since 1950, especially at the expense of arable land. The increase was largest in the regions 2, 4, 7, and 8 (Table 1) due to the growth of cities, such as Eindhoven and 's-Hertogenbosch. Within the category of arable land, figures were available about the production of so-called 'main crops', which together account for 83% of the total area of arable land in the province in 1950 and 68% in 2010. The information of crop type was not spatially explicit, meaning that we could not allocate crop types to their specific locations within each district. The share of corn increased significantly from 1950 to 2010, at the expense of grain and, to a lesser extent, of grassland (Table 2).

Crop yields increased significantly (Table 3). The average yield of grain in 2010 was 3.3 times as high as in 1950. The yield of potato and sugar beet approximately doubled. The yield of grassland increased by one quarter.

**Table 1.** Total area and division in main categories of land use of the regions of Figure 2. Source: CBS, 2010.

| Region | Area (ha) | Division (%) | | | | | | | |
|---|---|---|---|---|---|---|---|---|---|
| | | Arable | | Urban | | Forest & Nature | | Other | |
| | | 1950 | 2010 | 1950 | 2010 | 1950 | 2010 | 1950 | 2010 |
| 1 | 31,299 | 80 | 70 | 2 | 11 | 4 | 4 | 14 | 15 |
| 2 | 22,213 | 74 | 62 | 6 | 17 | 15 | 14 | 6 | 7 |
| 3 | 12,191 | 56 | 51 | 3 | 8 | 11 | 10 | 30 | 32 |
| 4 | 37,900 | 74 | 61 | 9 | 25 | 8 | 8 | 9 | 6 |
| 5 | 58,087 | 74 | 75 | 4 | 12 | 13 | 12 | 9 | 1 |
| 6 | 73,774 | 65 | 55 | 6 | 19 | 16 | 15 | 13 | 11 |
| 7 | 85,178 | 68 | 50 | 9 | 26 | 23 | 21 | 0 | 3 |
| 8 | 119,630 | 74 | 59 | 4 | 15 | 21 | 20 | 0 | 6 |
| 9 | 64,935 | 68 | 67 | 3 | 8 | 26 | 25 | 3 | 0 |
| Total | 505,208 | 71 | 61 | 5 | 17 | 18 | 17 | 6 | 6 |

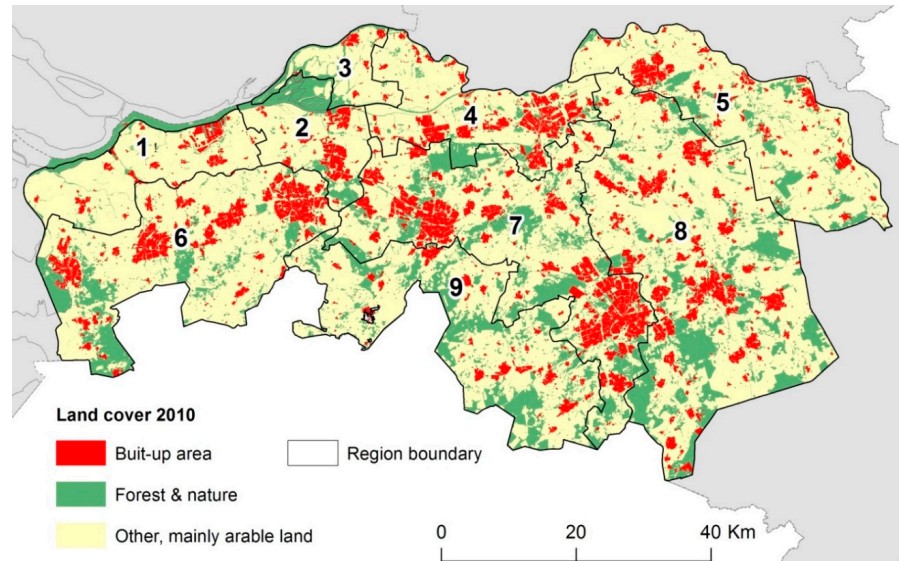

**Figure 2.** Main categories of land use in 2010 and division of the province of Noord-Brabant into regions for which CBS Statistics issued figures about crop production (Table 1). Source: CBS, 2010.

**Table 2.** Area division of main crops in the province of Noord-Brabant.

| | Division (%) | |
|---|---|---|
| | 1950 | 2010 |
| Cereal | 33 | 11 |
| Potato | 7 | 8 |
| Beet | 7 | 4 |
| Corn | 1 | 27 |
| Grass | 51 | 49 |

**Table 3.** Annual harvest ($Y$) in metric ton (1000 kg) product per ha, excluding straw and foliage. Cereal, potato, sugar beet, and fodder beet in ton fresh matter; corn and grass in ton dry matter.

| Region | Cereal | | Potato | | Sugar Beet | | Fodder Beat | | Corn | | Grass | |
|---|---|---|---|---|---|---|---|---|---|---|---|---|
| | 1950 | 2010 | 1950 | 2010 | 1950 | 2010 | 1950 | 2010 | 1950 | 2010 | 1950 | 2010 |
| 1 | 3.4 | 9.1 | 24.0 | 49.0 | 42.3 | 81.7 | 57.0 | - | 9.0 | 17.1 | 9.0 | 11.5 |
| 2 | 2.8 | 9.4 | 23.0 | 50.6 | 39.2 | 82.0 | 54.8 | - | 9.0 | 18.1 | 9.0 | 11.5 |
| 3 | 3.2 | 9.3 | 23.1 | 50.7 | 41.4 | 79.0 | 55.8 | - | 9.0 | 17.3 | 9.0 | 11.5 |
| 4 | 2.5 | 8.2 | 21.5 | 50.3 | 35.8 | 81.3 | 53.6 | - | 9.0 | 16.7 | 9.0 | 11.5 |
| 5 | 2.4 | 6.4 | 22.0 | 52.2 | 32.9 | 74.5 | 53.0 | - | 9.0 | 17.1 | 9.0 | 11.5 |
| 6 | 2.4 | 8.5 | 21.4 | 46.7 | 34.0 | 74.6 | 49.6 | - | 9.0 | 16.5 | 9.0 | 11.5 |
| 7 | 2.5 | 6.1 | 22.2 | 49.5 | 30.2 | 74.9 | 52.1 | - | 9.0 | 16.2 | 9.0 | 11.5 |
| 8 | 2.2 | 6.3 | 1.9 | 53.9 | 29.9 | 73.5 | 50.9 | - | 9.0 | 15.9 | 9.0 | 11.5 |
| 9 | 1.8 | 7.2 | 22.3 | 50.4 | 29.7 | 73.4 | 50.2 | - | 9.0 | 15.8 | 9.0 | 11.5 |
| Average | 2.4 | 7.8 | 22.4 | 50.4 | 38.7 | 76.9 | 51.6 | - | 9.0 | 16.3 | 9.0 | 11.5 |

*2.3. Estimating Changes in Annual Groundwater Recharge*

2.3.1. Groundwater Recharge of Urban Area

On the basis of an international literature review, eight water balance studies of urban districts in the Netherlands, and figures about the percentages of pavement in Dutch cities, De Graaf et al. [15] conclude that Dutch cities contribute less to the groundwater recharge than the rural area of the Netherlands. They estimate the evaporation of an average city in the Netherlands, including surface water and vegetated areas, at 66% of $E_{ref}$, while on average, 25% of the precipitation is discharged through the sewer system. According to the authors, wastewater discharges are "approximately" equal to the supply of drinking water and industrial water to the city (to support this: 99.5% of the Dutch population is connected to the sewer system [16], 0.5% tap water is used for gardening and car washing [17], and sewer leakages in Noord-Brabant are 2.5% [18]). This leads to the following approach of the groundwater recharge in urban areas:

$$R = 0.75P - 0.66E_{ref} \tag{1}$$

This simple approximation ignores differences in the fraction of paved area and the technology of sewer systems between 1950 and 2010. In 1950, fewer buildings were connected to the sewer system, and towns and cities were less densely urbanized, which is beneficial for groundwater recharge. On the other hand, the sewers were of lower quality so that they presumably sometimes drained groundwater. In addition, over the past few decades, the roofs of new buildings were less often connected to the sewer system but used wadi systems to infiltrate excess rainwater. Since we do not have sufficient information to judge whether the urban category in 1950 generated more or less groundwater recharge than in 2010, we applied Equation (1) to both periods.

2.3.2. Groundwater Recharge of Nature

Within the category 'nature', we could only distinguish the subcategories 'dry open areas', 'wet open areas', and 'forests' for both periods, since the topographic map of 1950 did not allow more detail. We used an actual evapotranspiration ($E$) of 400, 500 and 600 mm·y$^{-1}$, respectively, for these subcategories, based on literature [19–21]. Since surface runoff hardly occurs in the Netherlands, we computed groundwater recharge for nature as:

$$R = P - E \tag{2}$$

This is a rough but acceptable estimate, given the small share of nature areas within the province and the substantial lack of reliable evaporation measurements in nature areas.

2.3.3. Groundwater Recharge of Arable Land

For their photosynthesis, plants absorb $CO_2$ from the atmosphere by diffusion through their stomata. During this $CO_2$ uptake, water vapor diffuses from the substomatal cavities to the atmosphere, i.e., transpiration ($T$). Crop yield $Y$ is therefore linearly correlated to transpiration $T$ [22]. We used this principle to convert published data on the annual yield of harvested products (Table 3) into figures on the annual actual evapotranspiration, $E$. We did this in three steps:

First, we calculated the dry-matter production of the total annual biomass, excluding roots, by multiplying published crop yields ($Y$) (Table 3) with a dry weight coefficient ($C_{DM}$) we took from [23,24] (Table 4). $C_{DM}$ is equal to 1.0 for corn and grass because their yields are expressed as total dry-matter production in Table 3. The yield of cereal in Table 3 does not include straw, which results in $C_{DM}$ greater than 1.0 to obtain the total annual biomass. The harvest of potato and beet (sugar beet and fodder beet) is expressed as fresh weight in Table 3, so that their dry-matter production is calculated with a coefficient $C_{DM} < 1$.

**Table 4.** Parameter values of Equation (4) to convert annual harvest *Y* (Table 3) into annual groundwater recharge *R*.

|  | $C_{DM}$ (kg Dry Matter/kg Product) | $C_T$ (kg H$_2$O/kg Dry Matter) | $(E_s + E_i)$ (mm·y$^{-1}$) |
|---|---|---|---|
| Cereal | 1.67 | 300 | 117 |
| Corn | 1.00 | 210 | 127 |
| Beet | 0.23 | 230 | 115 |
| Potato | 0.25 | 290 | 115 |
| Grass | 1.00 | 370 | 133 |

Second, we calculated the transpiration (*T*) of each crop by multiplying *Y* with a transpiration coefficient ($C_T$). This is the reciprocal of the water-use efficiency of productivity, which indicates how many kilograms of water is needed for the production of one kilogram of dry matter. The annual transpiration thus becomes:

$$T = 10^{-4}C_{DM}C_T Y \tag{3}$$

where *T* = transpiration rate (mm·y$^{-1}$), $C_{DM}$ = dry matter coefficient (kg dry matter/kg harvested product), $C_T$ = transpiration coefficient (kg H$_2$O/kg dry matter), *Y* = crop yield provided by CBS Statistics (kg ha$^{-1}$·y$^{-1}$). The values for $C_T$ in Table 4 were also taken from [23,24].

Third, we accounted for evaporation from the soil (soil evaporation, $E_s$) and from the wet leaf surface during and after a precipitation event (interception evaporation, $E_i$), which occurs in addition to evaporation by transpiration. We assumed that the sum of both terms ($E_s + E_i$) is constant: When there is a lot of bare soil, $E_i$ is low and $E_s$ high, but when the soil is well covered, $E_i$ is high and $E_s$ low. We verified this assumption with the soil–water–atmosphere–plant model SWAP [25] for a loamy fine sand (most common soil type in Noord-Brabant) with an optimal groundwater regime for crops and the daily meteorological data described in Section 2.1 (further details in [13]). SWAP is a physically-based one-dimensional model with a vertical discretization of 1–5 cm and a time-step of 1 d, which mimics vertical transport of water in the unsaturated zone, as well as all the actual evaporation terms (*T*, $E_s$, and $E_i$). We simulated the sum for the most important agricultural crops and found a typical value of 125 mm·y$^{-1}$, with a range from 115 mm·y$^{-1}$ for potato and beet up to 133 mm·y$^{-1}$ for grass (Table 4).

The final groundwater recharge on a yearly basis thus becomes as follows:

$$R = P - (T + (E_s + E_i)) = P - (10^{-4}C_{DM}C_T Y + (E_s + E_i)) \tag{4}$$

The annual figures for *R* of were calculated for each region of Figure 2 and assigned to all arable land within that region.

*2.4. Modelling the Effect on Groundwater Tables*

We applied a distributed groundwater model to analyze how changes in *R* might have influenced groundwater levels. This model [26] is based on a regional hydrological and hydrogeological database built and maintained by the regional water boards, the provincial authorities, and the provincial drinking water company. It uses MODFLOW as a simulation program [27] and has a spatial resolution of 250 m and a temporal one of 5 d. Drainage of groundwater to surface waters is a non-linear function of groundwater level relative to surface water level. The model contains the groundwater abstractions (total 340 million m$^3$·y$^{-1}$), surface water system, and irrigation and drainage system of 2010. It has been calibrated using measured heads and measured surface water discharges. The head residuals in the shallow groundwater are in the order of 5 to 20 cm, with a proper reproduction of the seasonal variation. The deviation from the measured fluxes is less than 15 % [26]. Given the resolution and the size of the model area, this makes the model suitable for the simulation of the impact of recharge variations.

To apply the model we scaled annual figures of *E*-nature and *T*-agriculture from respectively Equations (2) and (3) to 5-day values using transfer functions that we derived with the simulation program for actual evapotranspiration and unsaturated groundwater flow SWAP [25] (Figure 3). The 5-d values of ($E_s + E_i$) for different crops were also simulated using SWAP. We imposed the 5-d groundwater recharge obtained this way as an upper boundary condition to the groundwater model. We did this for the recharge of both 1950 and 2010. The result is a calculated change in groundwater level resulting from a change in groundwater recharge.

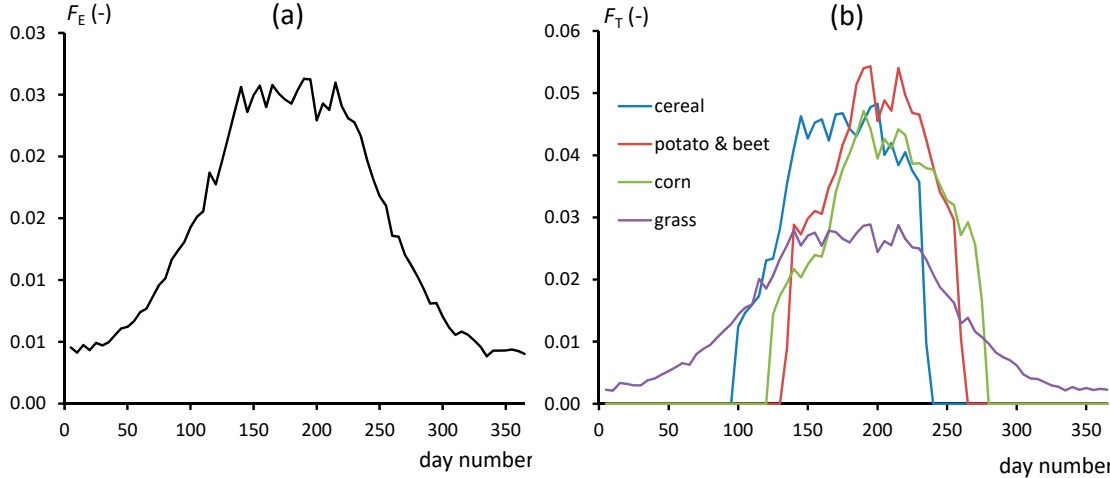

**Figure 3.** Multiplication factors to scale annual evapotranspiration figures for nature areas (**a**) and annual transpiration figures of major crops (**b**) to 5-day values. Based on simulations with the soil–water–atmosphere–plant model (SWAP; 2007–2013), using standard crop parameter values published in [28].

The simulated effects of interventions in water management may depend on the amount of groundwater recharge. To illustrate this phenomenon, we simulated the effect of current (2010) groundwater abstractions on groundwater tables twice: once by imposing the groundwater recharge of 1950 on the model, and once using the recharge of 2010.

## 3. Results

### 3.1. Changes in Annual Groundwater Recharge

The provincial averages (2007–2013) of the precipitation $P = 831$ mm·y$^{-1}$ and reference evaporation $E_{ref} = 601$ mm·y$^{-1}$ result, according to Equation (1), in a groundwater recharge rate in the category 'urban' of $R = 226$ mm·y$^{-1}$. This number, which applies to both 1950 and 2010, is about the same as the potential precipitation surplus of $P - E_{ref} = 230$ mm·y$^{-1}$. This means that compared to arable land the urban recharge is relatively low because in summer, the actual evapotranspiration of arable land is often reduced due to drought and crop damage: Urbanization in the Netherlands in general leads to reduction of the recharge.

For nature areas, we found a decrease of groundwater recharge from 238 to 202 mm·y$^{-1}$, which is due to the expansion of forest.

Table 5 presents the annual recharge for arable land obtained from Equation (4) using the crop yields $Y$ of Table 3, the parameter values of Table 4, and the average precipitation for the period 2007–2013 of $P = 831$ mm·y$^{-1}$. Taking land-use areas into account, the provincial average recharge in 1950 is equal to $R = 429$ mm·y$^{-1}$ and $R = 294$ mm y$^{-1}$ in 2010. The decrease in recharge from 1950 to 2010 is largest—157 mm y$^{-1}$—in the category 'arable land', mainly due to increased crop production (Table 5). That the biggest change occurred in arable land does not mean that this category also contributed most to the average drop in groundwater recharge throughout the province,

since urbanization from 1950 to 2010 led to less arable land (Table 1). If we express how much each category contributed to the area-weighted average decline, then 'arable land' and 'urban area' have about the same impact.

**Table 5.** Average groundwater recharge *R* in Noord-Brabant of urban area, nature, and main agricultural crops in 1950 and in 2010 according to Equation (4). The category 'arable' gives the area-weighted average of cereal, corn, beet potato, and grassland.

|  | $R$ (mm·y$^{-1}$) | | Decline in $R$ | |
| --- | --- | --- | --- | --- |
|  | **1950** | **2010** | **mm·y$^{-1}$** | **%** |
| Urban area | 226 | 226 | 0 | 0 |
| Nature | 238 | 202 | 36 | 15 |
| Cereal | 594 | 321 | 273 | 46 |
| Corn | 515 | 361 | 154 | 30 |
| Beet | 511 | 309 | 202 | 40 |
| Potato | 554 | 350 | 203 | 37 |
| Grassland | 365 | 272 | 93 | 25 |
| Arable | 521 | 364 | 157 | 30 |

### 3.2. Changes in Groundwater Tables

The groundwater model has been run with the transient recharge of 1950 and with that of 2010. Figure 4 shows the average difference in the piezometric head in the topmost aquifer. The largest fall in groundwater level from 1950 to 2010 due to land-use changes and changes in crop yield is simulated in areas with little surface water. The average decline in the regions with a sandy soil (regions 6, 7, 8, and 9) is 33 cm while the average for the entire province is 27 cm. The decrease in groundwater recharge is most pronounced in the growing season and therefore causes a larger drop in low groundwater levels than in high groundwater levels (results can be found in [13]).

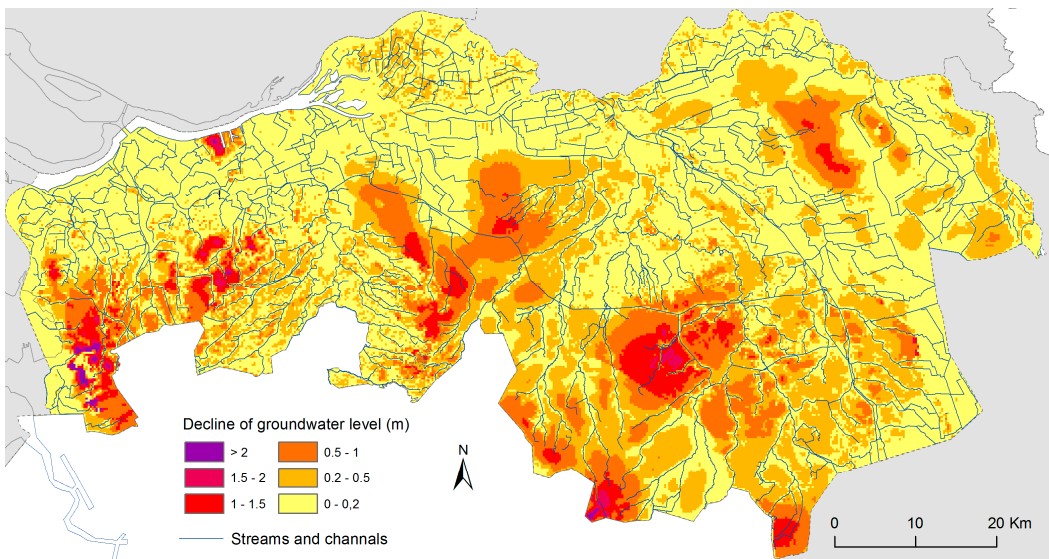

**Figure 4.** Simulated decline in groundwater level due to change in land use and increase of crop yields between 1950 and 2010.

### 3.3. Effect of Land-Use and Crop Yield on Simulated Effects of Groundwater Abstraction

The average drawdown caused by the groundwater extractions of 2010 was 10 cm for the larger groundwater recharge of 1950, while it was 15 cm with the lower groundwater recharge of 2010 (Figure 5). The median drawdown was 3 cm for 1950 and 4 cm for 2010, which is statistically more informative for the impact because of the large drawdown in the immediate vicinity of the extractions.

The impact of the abstractions thus increases with decreasing groundwater recharge. In other words, the drawdown of a constant extraction grows as crop production in agriculture increases and the urban area expands. This is due to the non-linear behavior of the top-system caused by the groundwater level dropping below drains and the shift from drainage to infiltration. The latter is associated with higher resistances than the former [29].

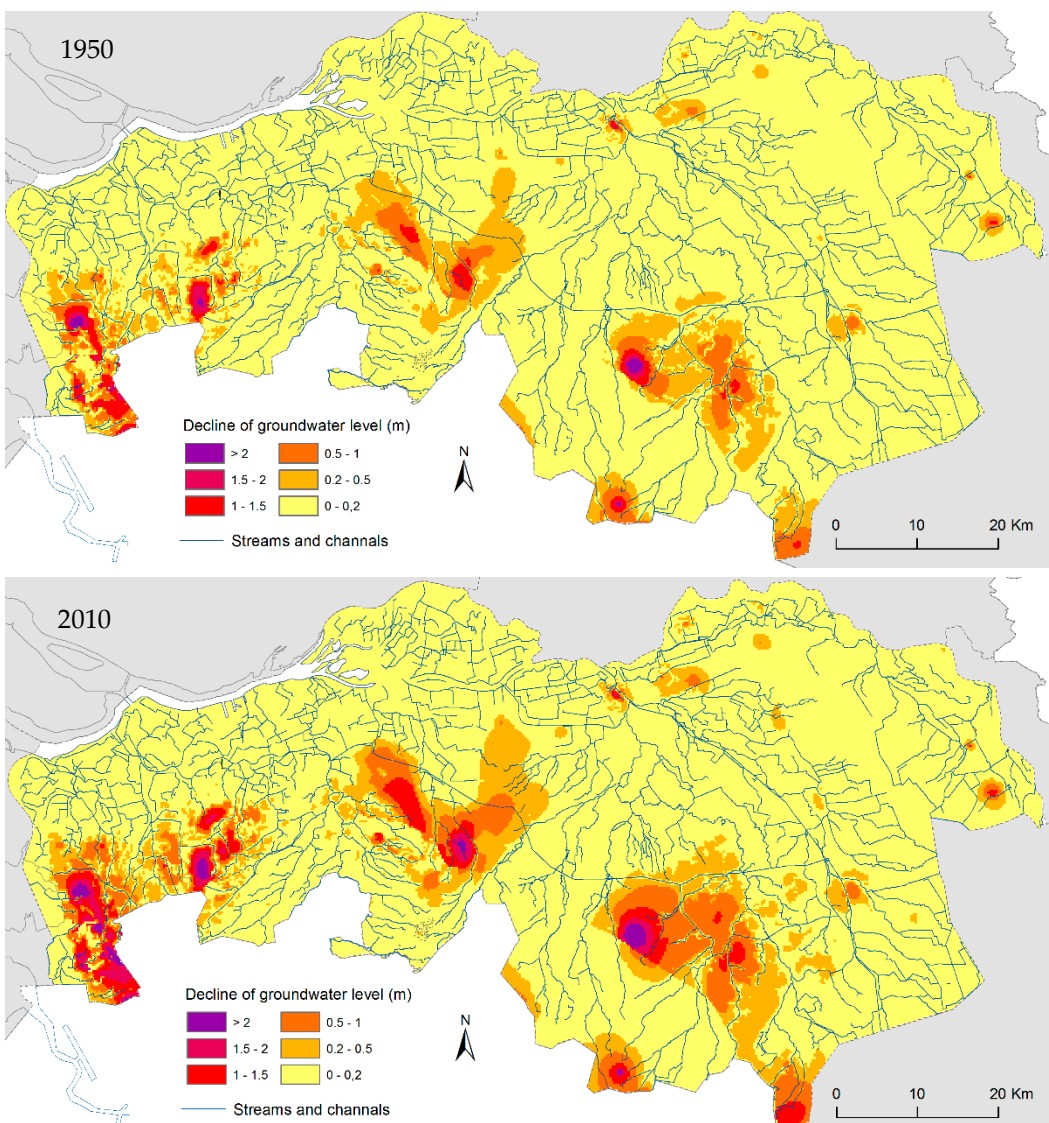

**Figure 5.** Decline of the groundwater table by current groundwater abstractions, simulated with the groundwater recharge of 1950 (**above**) and of 2010 (**below**).

## 4. Discussion

### 4.1. Limitations of Our Study

In this study, we dealt with the consequences of changes in land use and in crop yield on the groundwater levels in the province of Noord-Brabant. We did not take the land improvements that were carried out on a large scale after World War II into account explicitly. The main aim of these improvements was to prevent flooding and lower the groundwater table in the spring so that the soil could be cultivated with heavier machines, and the soil temperature and moisture content allowed better growth earlier in the year. The effect of land improvement is implicitly included in our results as far as it leads to higher crop yields.

In 1950, there was no overhead irrigation while the current irrigation in the province amounts to 70 million m$^3\cdot$y$^{-1}$, equaling a water depth of 14 mm, which is 6% of the annual precipitation surplus ($P - E_{\text{ref}}$). Virtually all irrigation water stems from groundwater and is applied to the fields using sprinkler installations. The irrigation allows for extra plant growth, and the corresponding transpiration is accounted for in the calculation. The irrigation return flow does recharge the groundwater again, so the net loss of groundwater due to the irrigation is equal to the extra evapotranspiration. The transpiration part is accounted for in the recharge reduction calculated from the crop yield. The calculations do neglect the extra evaporation which occurs due to the sprinkling, but the remaining effects of the irrigation are taken into account implicitly.

The consequences of the expansion of paved surfaces and of the increase of shrubs and trees in nature reserves could not be properly quantified in this study. The recharge in urban areas is uncertain, because little is known about the evaporation of this category [30].

Our calculation of groundwater recharge for arable land neglects the increase in the efficiency of crop production. The harvest losses were larger in 1950 and there were more weeds on agricultural fields. By way of uncertainty analysis we therefore also analyzed a scenario in which the increase in transpiration of arable land between 1950 and 2010 is half as large as we deduced from the increase in crop production. This resulted in an average recharge in 1950 of $R$ = 370 mm$\cdot$y$^{-1}$ (instead of 294 mm$\cdot$y$^{-1}$). This scenario was thoroughly discussed with the eight members of the study commission on 'background decline', mentioned in Section 1. They agreed the resulting recharge in 1950 should be considered as an upper limit (details in [13]). The corresponding average decline in groundwater table in the sandy regions of was 23 cm (instead of 33 cm). In conclusion, our study suggests that the effect of changes in land use and crop production in 60 years' time lies approximately between 0.2 and 0.3 m.

*4.2. Comparison with Previous Work*

Van Bakel and De Wit [31] studied the effect of increased crop yield on a potato field in a fertile agricultural polder in the Netherlands. The controlled water level in the polder was optimized to crop growth. Using a model for water flow and crop growth, SWACROP [32], they simulated the effect on evapotranspiration of an observed 40% increase in potato yield in the period of 1955–1987 (33 y). They used three scenarios, which resulted in an increase in actual evapotranspiration of 76, 74, and 43 mm$\cdot$y$^{-1}$. The results of the first two scenarios were comparable to the 69 mm$\cdot$y$^{-1}$ they obtained from a water balance evaluation. When we linearly extrapolate this figure to the 60-year period of our study, we arrive at an evapotranspiration increase for the potato field of 125 mm$\cdot$y$^{-1}$. In our study, we found an increase in the evapotranspiration of potato of 203 mm$\cdot$y$^{-1}$ (Equation (4)). The larger increase in evapotranspiration for Noord-Brabant is reasonable, since the potato field of Van Bakel and De Wit [31] was designed in an optimal manner from the start, while the water management in the province of Noord-Brabant has been improved substantially for agriculture.

The average simulated decline of 0.2–0.3 m in 60 years' time (1950–2010) in the sandy regions of the province of Noord-Brabant is comparable to the decline since 1950 that Rolf [9] found in the analysis of piezometric time series in the Netherlands, away from groundwater extractions.

*4.3. General Implications*

We performed our analysis for the Dutch province of Noord-Brabant. We accounted for urbanization, increased agricultural crop yield, and vegetation changes in nature areas. These processes do occur in many other regions around the globe and will have a similar effect everywhere.

Our conclusion can also be generalized to say that the drawdown due to groundwater extractions may change over time and is ambiguous as long as the drainage situation and groundwater recharge are not properly defined. Generally, the effect of groundwater extraction increases with decreasing groundwater recharge. The non-linearity of groundwater systems becomes more important when longer time periods are considered. This means the effects of influences are interdependent and the order in time needs to be considered for long-term hydrological studies. This result may be of

importance to environmental impact assessment studies. Should, for instance, the compensation budget to farmers of a groundwater abstraction be based on the land use and crop yield in the year that the abstraction started or on the current land use and crop yield? This is a question with possibly large technical, juridical, and financial consequences.

To our knowledge, our study is the first example of forensic hydrology into anthropogenic causes of possible water scarcity in a humid region. We foresee that the importance of forensic hydrology will increase substantially in the coming decades. Facing climate change, a growing human population, and more welfare, the demand for water especially for agriculture, households, and industries will increase and, with that, the risk of conflicts as well. Some changes will be very gradual, as we demonstrated with the gradual increase in crop production and land-use in the province of Noord-Brabant. This emphasizes the need to properly register changes in the landscape and to perform sufficient hydrological measurements. Unlike our forensic study, current and coming changes in landscape patterns nowadays can be observed quite easily with the aid of satellites.

## 5. Conclusions

Our study is an example of forensic hydrology, a branch of science that we expect to gain importance the coming decades. Historical data about land use and crop yield are scarce or even lacking, which means that we will never be able to accurately assess the changes in groundwater recharge and groundwater levels that occurred between 1950 and 2010. Nevertheless, our results suggest that changes in land-use and crop yield reduced the recharge of groundwater, which in turn caused a decline of groundwater levels since the 1950s in the Dutch province of Noord-Brabant. In the sand landscape of the province, which resembles the sand landscape of the other high part of the Netherlands (Figure 1), the simulated decline is on average 0.2–0.3 m in 60 years' time (1950–2010). This means that we found a plausible cause which largely explains the background decline of groundwater levels that occurred in the sandy region of the Netherlands. More research is needed, however, to generalize our findings for other areas. Arable land showed the strongest reduction in groundwater recharge. However, because the urban area has increased at the expense of arable land, urbanization is also important to the decline of groundwater levels.

We demonstrated that the drawdown caused by groundwater abstractions depends on the amount of groundwater recharge related to land use and crop yield. This means that, to avoid ambiguous results, the entire hydrological history should considered when evaluating large scale impacts.

**Supplementary Materials:** The following are available online at http://www.mdpi.com/2073-4441/11/3/478/s1.

**Author Contributions:** J.-P.M.W. conceived the idea; J.-P.M.W. and W.J.Z. gathered and analyzed the data; J.-P.M.W. computed groundwater recharge rates; W.J.Z. simulated groundwater levels; J.-P.M.W. composed the draft; J.J.B. and W.J.Z. contributed to the interpretation of results and the revision of the draft; J.-P.M.W. managed the project for financial support.

**Funding:** This research received funding from the Dutch drinking water companies Vitens and Brabant Water, as well as by Vewin, the association of drinking water companies in the Netherlands.

**Acknowledgments:** We are also indebted to Frans Aarts and Jan van Bakel for the fruitful discussion we had, to Ruud Bartholomeus and two anonymous reviewers for their valuable comments on the manuscript, and to Gijsbert Cirkel, Inke Leunk and Bernard Raterman for their technical support.

**Conflicts of Interest:** The authors declare no conflict of interest.

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
