# Peer review of "Forensic Hydrology Reveals Why Groundwater Tables in The Province of Noord Brabant (The Netherlands) Dropped More Than Expected"

_water, doi:10.3390/w11030478_

Round 1
Reviewer 1 Report
Manuscript: Forensic hydrology reveals why groundwater tables in the Netherlands dropped more than expected
General remarks:
The authors present an interesting study about the analysis of reasons for a drop down of ground water levels between 1950 and 2010 in a province of Netherland. For that purpose, water balance calculations taking into account anthropogenic changes in groundwater recharge and rising crop yields were carried out using simple equations, as well as physically based simulation models such as SWAP and MODFLOW. The problem defined by the authors is the difference between measured and simulated drop down of groundwater levels described as the background decline. This background decline is defined as the fact that groundwater levels from 1950 to 1989 had dropped on average 0.3 m more as compared to calculations using time series of measured precipitation and reference evapotranspiration. In general, the study is well written and tackles an interesting part of hydrology, the so called forensic hydrology. However, the authors used a broad mixture of empirical equations such equation (1) and (2) as well as the simulation models SWAP and WOFOST. Due to this mixture, from my point of view, some essential details of the model combinations, data flow and model applications are missing. The following comments might help to improve the submitted manuscript for a final publication.
Introduction:
Excellent written chapter.
Comment to line 68 to 71: Interesting statement of Stromingen
Line 88-89: This means that previous model applications and calculations cited in the chapter Introduction have not taken into account these antropogenic changes in groundwater recharge ?.
Line 102-104: The period around 1950 corresponds to 1947-1953 and that around 2010 to 2007 to 2013 ?
Line 113-116: How are these nine regions defined in Figure 1? This information is difficult to found at the website cited in line 115.
Table 1-3: Data from CBS ?.
Line 139-140: These calculations of groundwater recharge using daily values of precipitation and reference evapotranspiration were also carried out with a daily time step ? However, equation (1)-(4) are obviously valid for yearly time steps. This contradiction should be explained in more detail by the authors.
Line 177-179: How are these subcategories defined and how is E calculated ?. A short additional information about that purpose might improve the manuscript.
Line 197-199: I have checked the validity of equation (3) with the following data:
Assumed yield of beet: 5000 kg ha-1, CDM=0.23; CT = 230. This resulted in a transpiration of 26.32 mm y-1, Is that correct ?.
Line 206-210: How were these data calculated by SWAP?, Which soil and crop data and meteorological data from 2007 to 2013 ?.
Line 229-235: In addition to the comments on Line 139-140, if the calculations of groundwater recharge were carried using daily values of precipitation and reference evapotranspiration, why is it necessary to scale E-nature and T-agriculture on 5-day values using SWAP, how is E-nature scaled by the application of SWAP ?. In the actual state of the manuscript, this is difficult to understand.
Fig. 2: If understand the contents of this figure correctly, Fig.2 a means a 5-daily value of E-nature calculated as E-nature_5day= Fe(Figure2a) *E-nature_year ?. This would lead to the following results: E-nature_year= 400 mm (see line 179) for day 200 with Fe=2.5 would result in a 5-day value of 900 mm ? Is that correct.
Final remarks:
The authors present an interesting and innovative study about forensic hydrology. However, the chapter Material and Methods has to be distinctly improved by a more detailed description of the applied methods. This statement is also justified by the fact that many references necessary for an understanding of the applied methods are written unfortunately all in Dutch. This is difficult for readers outside the Netherlands.
Author Response
General
Herewith we respond to the comments of reviewer 1 of our manuscript ‘Forensic hydrology reveals why groundwater tables in the Netherlands dropped more than expected’. We numbered each comment in ascending order. Changes in the manuscript are highlighted in yellow.
The most important comment of both reviewer 1 seems that the methods are still not well described. We tried to solve this problem by adding a new subsection ‘General approach’ in which we describe the connection between the different parts of our analysis, with special attention to the use of meteorological data.
We believe the review substantially helped us to improve the manuscript and thank the reviewer for his or her valuable comments. We hope that it is now acceptable for publication in WATER.
Reply to comments
1. Excellent written chapter.
Reply: We thank the reviewer for the appreciating words
2. Comment to line 68 to 71: Interesting statement of Stromingen
Reply: We thank the reviewer for the appreciating words
3. Line 88-89: This means that previous model applications and calculations cited in the chapter Introduction have not taken into account these antropogenic changes in groundwater recharge?.
Reply: Indeed, these changes have been neglected in previous studies, as we now explicitly mention in line 91-92 of the revised manuscript
4. Line 102-104: The period around 1950 corresponds to 1947-1953 and that around 2010 to 2007 to 2013?
Reply: Indeed: information added, line 113
5. Line 113-116: How are these nine regions defined in Figure 1? This information is difficult to found at the website cited in line 115.
Reply: Information added, see line 156-158.
6. Table 1-3: Data from CBS ?.
Reply: Information added, line 173
7. Line 139-140: These calculations of groundwater recharge using daily values of precipitation and reference evapotranspiration were also carried out with a daily time step ? However, equation (1)-(4) are obviously valid for yearly time steps. This contradiction should be explained in more detail by the authors.
Reply: We admit that the combination of annual figures about land use and crop yield with daily meteorological values can be confusing. For this reason we introduced a new subsection (2.1) in which we describe the general approach of the study and the use of meteorological data. We hope this will give the readers enough support to understand the rest of Section 2. See line 122-151.
8. Line 177-179: How are these subcategories defined and how is E calculated ?. A short additional information about that purpose might improve the manuscript.
Reply: Information added: line 201.
9. Line 197-199: I have checked the validity of equation (3) with the following data: Assumed yield of beet: 5000 kg ha-1, CDM=0.23; CT = 230. This resulted in a transpiration of 26.32 mm y-1, Is that correct ?
Reply: Almost, the exact number is 26.45 mm/y. Note that a yield of 5000 kg/ha is not very realistic: Table 3 provides yields for beet that are about ten times higher. 50,000 kg/ha would yield 265 mm/y.
10. Line 206-210: How were these data calculated by SWAP?, Which soil and crop data and meteorological data from 2007 to 2013 ?.
Reply: Information added: line 231-233.
11. Line 229-235: In addition to the comments on Line 139-140, if the calculations of groundwater recharge were carried using daily values of precipitation and reference evapotranspiration, why is it necessary to scale E-nature and T-agriculture on 5-day values using SWAP, how is E-nature scaled by the application of SWAP ?. In the actual state of the manuscript, this is difficult to understand.
Reply: We are sorry that our description of the method was unclear. We hope that subsection 2.1 (General approach) now gives enough support to avoid misinterpretations.
12. Fig. 2: If understand the contents of this figure correctly, Fig.2 a means a 5-daily value of E-nature calculated as E-nature_5day= Fe(Figure2a) *E-nature_year ?. This would lead to the following results: E-nature_year= 400 mm (see line 179) for day 200 with Fe=2.5 would result in a 5-day value of 900 mm ? Is that correct.
Reply: We thank the reviewer for his/her observation: the y-axis is too large a factor 100. We modified figure 2, see line 269.
13. Final remarks:
The authors present an interesting and innovative study about forensic hydrology. However, the chapter Material and Methods has to be distinctly improved by a more detailed description of the applied methods. This statement is also justified by the fact that many references necessary for an understanding of the applied methods are written unfortunately all in Dutch. This is difficult for readers outside the Netherlands.
Reply: We understand and thank Reviewer 1 for his or her valuable comments which, in our opinion, substantially helped to improve the manuscript
Reviewer 2 Report
Review of “Forensic hydrology reveals…” Witte et al.
This article studies the decline of groundwater in the Netherlands over the past half-century by compiling historical records of groundwater and using a common groundwater model. This manuscript is topically appropriate for the readership of Water and generally well written. My general / major comment is that the article appears to be more broadly focused that it actually is. I am recommending that the authors revise their manuscript before it can be considered suitable for publication. Comments are detailed below.
specific comments:
title, this study focuses on a detailed analysis of a small region within the Netherlands. There is nothing wrong with this, but the title should reflect the small scale study instead of implying that the authors study the entire country.
introduction, paragraph one is somewhat short, the authors may wish to expand this out in more detail
line 44, long paragraph that is more historical than motivation; consider revising.
line 108, I appreciate that the authors want to use a smaller region for ease of approach, but this sentence is somewhat opinionated for a scientific article. Also, this echo’s my major point above and suggestion that the authors embrace the smaller scale focus of this work.
line 109-110, this is pretty opinionated for a scientific article.
line 169 replace less with fewer
lines 176-180. Do you mean natural recharge? This seems to be very important, would vary spatially with soil type and also depend on runoff. Also, do you mean just E or ET? Seems this would vary with vegetation cover and soil type.
line 182- this seems pretty detailed given the approximate crop formulations used. It’s also out of synch with the other sections.
Line 216. Much of what is presented in the recharge above should be in MODFLOW as the Farm Package, etc. The authors might want to provide more detail and explain the reasoning for their assumptions.
lines 247-253. Are there estimates of recharge the authors can compare to? Certainly there are excellent observations of ET at the many long-running flux towers and energy balance sites around the Netherlands (e.g. Cabaw) that can provide some estimate of the applicability of the results shown here.
line 254, one sentence paragraph, consider revising.
lines 273+, are there observations of groundwater decline that can be compared to model results?
line 303. It seems that irrigation recharge and return flows are an important component of the water budget and may be very significant in the current study. The authors may want to justify this assumption in more detail.
line 353, I’m not sure the authors are the first to simulate hydrologic changes in a retrospective way. I may be missing a nuanced difference, but many groundwater studies have sought to reconstruct hydrologic impacts to groundwater.
line 363. I’m still not sure what the authors mean by Forensic Hydrology. If this is a major component of the contribution of this work it should be better clarified and defined. Otherwise I think the authors should delete.
Conclusions, general. I think the simulations suggest, but do not demonstrate. More work would be needed to meet this higher standard.
Author Response
General
Herewith we respond to the comments of reviewer 2 of our manuscript ‘Forensic hydrology reveals why groundwater tables in the Netherlands dropped more than expected’. We numbered each comment in ascending order. Changes in the manuscript are highlighted in yellow.
The most important comment of reviewer 2 seems that the methods are still not well described. We tried to solve this problem by adding a new subsection ‘General approach’ in which we describe the connection between the different parts of our analysis, with special attention to the use of meteorological data.
We believe the review substantially helped us to improve the manuscript and thank the reviewer for his or her valuable comments. We hope that it is now acceptable for publication in WATER.
Reply to comments
1. title, this study focuses on a detailed analysis of a small region within the Netherlands. There is nothing wrong with this, but the title should reflect the small scale study instead of implying that the authors study the entire country.
Reply: We changed the title to reflect that the simulation were carried out for the province of Noord-Brabant (which is one of the larger of the 12 provinces in the Netherlands). We also explained that the results of Noord-Brabant are also valid for other parts of the Netherlands. See line 118-120.
2. introduction, paragraph one is somewhat short, the authors may wish to expand this out in more detail
Reply: Paragraph has been expanded. See line 30-35.
3. line 44, long paragraph that is more historical than motivation; consider revising.
Reply: The aim of this paragraph is to show that the reason for our research originates from political choices made in the past, as well as from a current social debate. The historical development shaped the current situation and we believe it is necessary background, especially with the current attention for socio-hydrology.
4. line 108, I appreciate that the authors want to use a smaller region for ease of approach, but this sentence is somewhat opinionated for a scientific article. Also, this echo’s my major point above and suggestion that the authors embrace the smaller scale focus of this work.
Reply: Similar developments have taken place around the globe, but the global scale is not appropriate for such an analysis due to data limitations. Sufficient data was available at the regional scale of Noord-Brabant, a province that it is large enough to show general trends (at least for the Netherlands; in other areas the urbanisation may increase recharge instead of decreasing it) - also see reply on comment 1.
5. line 109-110, this is pretty opinionated for a scientific article.
Reply: We deleted this sentence.
6. line 169 replace less with fewer
Reply: Done, see line 192.
7. lines 176-180. Do you mean natural recharge? This seems to be very important, would vary spatially with soil type and also depend on runoff. Also, do you mean just E or ET? Seems this would vary with vegetation cover and soil type.
Reply: It is indeed evapotranspiration, for we used the symbol E (the symbol ET can be interpreted as ExT). We motivated the simplicity of our approach in line 205-206.
8. line 182- this seems pretty detailed given the approximate crop formulations used. It’s also out of synch with the other sections.
Reply: Subsection ‘groundwater recharge of arable land’ is best worked out, so it seems logical to explain why transpiration yield is directly proportional to crop yield.
9. Line 216. Much of what is presented in the recharge above should be in MODFLOW as the Farm Package, etc. The authors might want to provide more detail and explain the reasoning for their assumptions.
Reply: The Farm process package of MODFLOW can perform calculations of agricultural water demand and of surface water and/or groundwater extraction to fill this demand. The parametrisation for these calculations has to be supplied by the user and the calculation concept does not allow for changes due to cultivation of new varieties and improved agricultural practices. Furthermore, the package does not allow for inputting the crop yield and back-calculating the transpiration. Therefore, the use of the Farm process would not have improved the simulations.
10. lines 247-253. Are there estimates of recharge the authors can compare to? Certainly there are excellent observations of ET at the many long-running flux towers and energy balance sites around the Netherlands (e.g. Cabaw) that can provide some estimate of the applicability of the results shown here.
Reply: There are no independent measurements of recharge available which can help to validate the used relations between crop yield and transpiration or the estimate for urban recharge. The relations between crop yield and transpiration have been established in numerous experiments in which both transpiration and crop yield were carefully measured, often with the aid of lysimeters (literature reviews in De Wit (1958) and Aarts (2000)). The evapotranspiration estimates from stations like Cabauw do not cover enough crops for a sufficiently long period. Moreover, these stations are usually situated in grasslands with optimal sufficient water regime.
11. line 254, one sentence paragraph, consider revising.
Reply: Each paragraph in section 3.1 describes the changes of recharge in a particular land use category, i.e. Urban, Nature and Arable. For the second category, we only needed one sentence. It does not seem logical to combine this sentence with one of the other two categories.
12. lines 273+, are there observations of groundwater decline that can be compared to model results?
Reply: Groundwater decline cannot be measured. Differences between heads measured around 1950 and 2010 are not an appropriate proxy for the calculated groundwater decline, because the calculations exclude the difference in meteorology and in initial groundwater heads. In the past, time series models have been applied to separate head changes due to variation in reference evaporation and precipitation from anthropogenic influences, which we have referred to (e.g. references 8, 9, and 15).
13. line 303. It seems that irrigation recharge and return flows are an important component of the water budget and may be very significant in the current study. The authors may want to justify this assumption in more detail.
Reply: Text has been adapted to justify our approach in more detail, see line 336-342
14. line 353, I’m not sure the authors are the first to simulate hydrologic changes in a retrospective way. I may be missing a nuanced difference, but many groundwater studies have sought to reconstruct hydrologic impacts to groundwater.
Reply: We agree and have adapted the text into a valid claim, se e line 388-389.
15. line 363. I’m still not sure what the authors mean by Forensic Hydrology. If this is a major component of the contribution of this work it should be better clarified and defined. Otherwise I think the authors should delete.
Reply: Information added, line 102-108
16. Conclusions, general. I think the simulations suggest, but do not demonstrate. More work would be needed to meet this higher standard.
Reply: Agree, see line 354, 401 and 408
We thank Reviewer 2 for his or her valuable comments which, in our opinion, substantially helped to improve the manuscript.
Round 2
Reviewer 1 Report
All my comments were adressed by the authors.
Reviewer 2 Report
thank you for revising your manuscript, I think it is now acceptable for publication.